# Correlation between the Neuropathic PainDETECT Screening Questionnaire and Pain Intensity in Chronic Pain Patients

**DOI:** 10.3390/medicina57040353

**Published:** 2021-04-07

**Authors:** Sebastian Lukas König, Michal Prusak, Sibylle Pramhas, Marita Windpassinger

**Affiliations:** 1Department of Anesthesia, Intensive Care Medicine and Pain Medicine, Division of General Anesthesia and Intensive Care Medicine, Medical University Vienna, Spitalgasse 23, 1090 Vienna, Austria; sebastian.koenig@meduniwien.ac.at (S.L.K.); michal.prusak@meduniwien.ac.at (M.P.); 2Department of Anesthesia, Intensive Care Medicine and Pain Medicine, Division of Special Anesthesia and Pain Medicine, Medical University Vienna, Spitalgasse 23, 1090 Vienna, Austria; sibylle.pramhas@meduniwien.ac.at

**Keywords:** neuropathic pain, chronic pain, PainDETECT questionnaire, numeric rating scale, outpatients

## Abstract

*Background and Objectives*: Pain is a multidimensional phenomenon with a wide range regarding the location, intensity and quality. Patients with chronic pain, in particular those suffering from mixed pain, often present a special challenge. The PainDETECT questionnaire (PD-Q) is a screening instrument designed to classify whether a patient has neuropathic pain (NP), often rated as more distressing compared to nociceptive pain. The objective of this study was to investigate whether the PD-Q score correlates with pain intensity, measured with the numeric rating scale (NRS), in chronic pain patients in an outpatient setting. *Materials and Methods*: A questionnaire-based study was conducted to identify the associations between the unidimensional NRS scale for pain intensity and the PD-Q score for screening of an NP component in an outpatient setting. Participants were asked to fill in the questionnaire themselves. *Results*: One hundred seventy-six participants completed the PD-Q questionnaire and rated pain on the NRS scale at the baseline visit. The PD-Q and NRS scores significantly correlated at the baseline visit and the 1-month follow-up visit in chronic pain patients. The identification of a neuropathic component in chronic pain may permit more targeted and effective pain management. *Conclusions*: The findings of our questionnaire suggest that a significant proportion of chronic pain patients had manifested features of NP at the first visit to the outpatient clinic. The PD-Q is a useful screening tool to alert clinicians of NP that may need further diagnostic evaluation or therapeutic intervention and may also help to predict treatment response. Further research is needed to investigate if a correlation is predictive of treatment response when pain therapy targets NP.

## 1. Introduction

Pain represents a major clinical, social and economic problem. Chronic pain is a multidimensional phenomenon with a wide range regarding the location, intensity and quality. In particular, the group of patients suffering from chronic pain often present a special challenge for physicians specialized in the field of pain management.

Epidemiological surveys reported that a large proportion of chronic pain patients, particularly ones with neuropathic symptoms, is not treated appropriately [1,2,3]. This may be caused by a lack of diagnostic screening tools, limitations in diagnostic accuracy or insufficient knowledge about effective drugs and their appropriate use in pain therapy regimes.

Neuropathic pain (NP), often underdiagnosed and undertreated, is a frequent condition affecting 7–10% of the general population [4]. It is caused by a lesion or disease of the central or peripheral somatosensory nervous system [5]. Most patients suffering from NP report severe, chronic symptoms, often difficult to manage in conventional pain therapy. This has a considerable impact on quality of life, causing major suffering and disability. The extent and severity can vary between individuals in a cohort of patients suffering from the same underlying disease or neural lesions [5,6]. The management of NP is unsatisfactory both in preventing its development and in halting or modifying its progression [7]. A recent study by Smith and colleagues reported that some classically “non-neuropathic” painful conditions can give rise to symptoms more commonly associated with NP and that these symptoms respond to anti-neuropathic medication [2]. Pain with a neuropathic character was found in 23% of patients with knee or hip osteoarthritis and was found 6 times more often in patients after knee surgery [8,9]. Research in patients with fibromyalgia reported a prevalence of neuropathic symptoms in 34% of patients [8].

It should be noted that not all patients affected by neural disorders or lesions develop neuropathic sensations [2]. Indeed, only 21% of patients suffering from herpes zoster infection develop NP [2]. These study results represent the fact that chronic pain with a neuropathic component is clinically difficult to diagnose and treat and requires a different diagnostic and therapeutic regime.

The common definition of the term “mixed pain” is a pain with an overlap of nociceptive and neuropathic symptoms [10,11]. It is applied for specific clinical diagnosis, such as low back pain or cancer pain, in which an overlap of the different pain types, nociceptive, neuropathic or nociplastic, in any combination, acting simultaneously or concurrently, is observed [11]. The diagnosis of mixed pain is made based on medical history-taking and clinical judgment, rather than by screening or diagnostic criteria, which is problematic for physicians in pain management [11,12]. The detection and targeted pain treatment in an early stage of disease course play a crucial role in the multimodal pain therapy regime.

Therefore, to identify chronic pain patients with potential mixed pain, validated screening tools such as the PainDETECT questionnaire (PD-Q) may be advantageous in the evaluation and management of complex pain.

The PainDETECT screening questionnaire (PD-Q) was designed to measure neuropathic components in pain. It has an easy-to-use paper and pencil version that can be performed in less than 5 min [13]. Studies have demonstrated that the PD-Q can be used to reliably distinguish the severity of pain in patients with NP [14].

Using the PD-Q to screen a patient with chronic pain, particularly one suffering from pain with both nociceptive and neuropathic components, allows the physicians to use targeted treatment according to the evaluation scores.

This study was designed to investigate whether the PD-Q score correlates with the corresponding numeric rating scale (NRS) score in chronic pain patients with mixed diagnoses in painful conditions in an outpatient setting.

## 2. Materials and Methods

### 2.1. Study Participants

Study subjects were recruited from outpatients diagnosed with chronic pain at the Division of Special Anesthesia and Pain Medicine at the Department of Anesthesia, Intensive Care Medicine and Pain Medicine, Medical University Vienna between September 2010 and June 2011. Patients were selected according to the following inclusion criteria: age ≥18 years and painful disease lasting for more than 6 months. Exclusion criteria were inability to perform PD-Q questionnaire; history of neurological or psychiatric diseases or disorders; abuse of medication, alcohol or drugs; or an existing therapy with WHO III analgesics. The study protocol was approved by the local Ethics Committee of the Medical University of Vienna. Informed consent was obtained from each participant.

### 2.2. Methods

Participants completed a validated self-administered questionnaire (PD-Q) and rated their pain on a numeric rating scale (NRS) at baseline visit and at the 1-month follow-up.

The PD-Q is a symptom-based assessment tool performed by clinicians to specify the subjective neuropathic pain experience of patients. With the score, pain can be classified into neuropathic, unclear or nociceptive pain groups. It was developed for the identification of a neuropathic component. It evaluates pain intensity, pain pattern and pain quality, with a sensitivity and specificity of over 80% [15]. The overall score ranges between –1 and +38, with higher scores indicating more likely NP (≤12 unlikely; 13–18 uncertain; ≥19 likely). The PD-Q is short, easy to understand and takes only a few minutes to complete and score; therefore, it is an optimal tool for time-poor clinicians. It can be performed for a wide range of patients with a potential neuropathic sensation in painful diseases and gives an accurate reflection of the patient’s neuropathic state.

Pain intensity was observed using the 11-point numeric rating scale (NRS; 0 = no pain; 10 = the worst pain). Participants were asked about their pain experienced at the moment of testing (current pain), the average pain during the last 4 weeks and the maximum pain during the last 4 weeks. The NRS score was evaluated at baseline visit and re-evaluated at the 1-month follow-up.

### 2.3. Statistical Analysis

Data were analyzed using IBM SPSS Statistics 27.0 (SPSS Version 23.0, IBM Corp., Armonk, NY, USA). Age and gender were analyzed using the appropriate descriptive statistics in the overall study population and all subgroups. To test for statistical differences in age between the subgroups, a *t*-test for independent samples and the Kruskal–Wallis test were used. The chi-squared test was used to screen for correlations between gender and the distribution into the subgroups. Participants were divided post hoc into subgroups based on PD-Q score (≤12, 13–18, ≥19) and anti-neuropathic therapy (with, without). We calculated descriptive statistics (arithmetic mean, standard deviation, minimum and maximum) to describe the findings of both the PD-Q and the three NRS scores at the two points in time and for all subgroups defined according to the PD-Q score and anti-neuropathic medication intake. Spearman’s rank correlation coefficient was used to search for correlations between the PD-Q scores and the individual NRS scores in both the overall study population and subgroups defined by PD-Q score and anti-neuropathic therapy. Correlations were calculated for baseline and follow-up visit. To test for a difference in PD-Q score and NRS between the groups with and without anti-neuropathic medication, we used the Mann–Whitney U test for independent samples at baseline and follow-up visit.

## 3. Results

In total, 176 patients filled in the PD-Q on the first outpatient visit (baseline). All subjects had a history of chronic pain sensations of at least 6 months.

The included patients had various underlying diagnoses, which for convenience and simplicity were allocated into 12 groups following the ICD-10 code. ICD-10 groups and the number of patients with the respective diagnoses are listed in Table 1.

The average age of participants at baseline visit was 57.6 years, with the oldest patient being 92 years old and the youngest being 27 years old. One hundred fifteen participants (65.3%) were women and 61 (34.7%) were men.

The mean ages were 54.5 years (SD 14.0, min 27.3, max 88.1), 59.4 years (SD 14.4, min 32.0, max 86.5) and 59.3 years (SD 17.5, min 27.8, max 92.7) in the groups of patients with likely, uncertain and unlikely NP, respectively. There was no statistically significant difference in age between the three groups.

According to the PD-Q score, at the baseline visit NP was likely (score ≥ 19) in 62 (35.2%), possible (score ≥ 13 to ≤18) in 55 (31.3%), and unlikely (score ≤ 12) in 59 (33.5%) patients. Of the group of patients with likely NP, 46 (74.2%) were women and 16 (25.8%) were men. Of the group with uncertain NP, 35 (63.6%) were women and 20 (36.4%) were men. Of the 59 patients with unlikely NP, 34 (57.6%) were women and 25 (42.4%) were men. There was no statistically significant correlation between the gender of the patients and the distribution into the three groups.

The mean PD-Q score at the baseline visit was 15.69 (SD 7.49, min 0, max 35). The mean NRS score at baseline visit was 6.86 (SD 2.62, min 0, max 10), the mean NRS of the most severe pain during the last 4 weeks was 8.15 (SD 2.23, min 0, max 10) and the mean NRS of average pain during the last 4 weeks was 6.74 (SD 2.62, min 0, max 10.0).

There was a significant positive correlation between the PD-Q score at the baseline visit and the current NRS score (pc = 0.21, *p* = 0.007), NRS of most severe pain during the last 4 weeks (pc = 0.2, *p* = 0.01) and NRS of average pain in the last 4 weeks (pc = 0.26, *p* < 0.001). When testing the three different groups of patients defined by likely, uncertain and unlikely NP, for a correlation between the PD-Q score and the three NRS scores, we found a significant positive correlation (pc = 0.4, *p* = 0.01) between the NRS of average pain in the last 4 weeks and the PD-Q score at the baseline visit in the group of patients with likely NP.

Of all 176 participants, 79 (44.8%) were prescribed anti-neuropathic medication (new onset or ongoing). The mean age of the participants who were prescribed anti-neuropathic medication was 58.46 years (SD 13.28, min 27, max 88). These 79 patients were divided into 48 (60.8%) women and 31 (39.2%) men. Of these, 30 were in the group with likely, 30 in the group with uncertain and 19 were in the group with unlikely NP.

In the group of patients under anti-neuropathic therapy, we found no significant correlation between the three NRS subgroup scores and the PD-Q score at the baseline visit.

Of the 97 patients without anti-neuropathic therapy, 32 had likely, 25 had uncertain and 40 had unlikely NP. The mean age in this group was 56.98 years (SD 13.28, min 27, max 92). Of the 97 patients in this group, 67 (69.1%) were women and 30 (30.9%) were men. A positive correlation was found in this group of patients between the PD-Q score and the current NRS score (pc = 0.21, *p* = 0.039), NRS of most severe pain during the last 4 weeks (pc = 0.26, *p* = 0.012) and NRS of average pain in the last 4 weeks (pc = 0.3, *p* = 0.003) at baseline.

The mean PD-Q score at baseline in the subgroup of patients with anti-neuropathic medication was 17.31 (SD 6.78, min 1, max 35), the mean NRS at the baseline visit was 7.01 (SD 2.46, min 0, max 10), the mean NRS of average pain during the last 4 weeks was 6.77 (SD 2.6, min 0, max 10) and the NRS of the highest pain in the last 4 weeks was 8.05 (SD 2.33, min 0, max 10).

In the group without anti-neuropathic medication, the mean PD-Q score at baseline was 14.75 (SD 7.76, min 0, max 34), the mean NRS at baseline was 6.74 (SD 2.75, min 0, max 10), the mean NRS of average pain during the last 4 weeks was 6.71 (SD 2.67, min 0, max 10) and the NRS of the highest pain in the last 4 weeks was 8.24 (SD 2.15, min 0, max 10).

The PD-Q score was significantly higher (mean difference 2.55, *p* = 0.021) in the group with anti-neuropathic medication than in the group without anti-neuropathic medication. Figure 1 shows the distribution of both the PD-Q score and the NRS score among the patients with and without anti-neuropathic medication.

Sixty-two patients had a follow-up visit after 1 month. Of these, 40 (64.5%) were female and 22 (35.5%) were male. The mean age in this group was 59.1 years (SD 16.7, min 27.3, max 88.6). Following the PD-Q score, NP was likely in 20 (32.3%), uncertain in 22 (35.5%) and unlikely in 20 (32.3%) patients at the 1-month follow-up. In the group with likely NP, 12 patients (60%) were women and 8 patients (40%) were men. Of the 20 patients with uncertain NP, 11 (57.9%) were women and 8 (42.1%) were men. Seventeen patients (73.9%) with unlikely NP were women and 6 patients (26.1%) with unlikely NP were men. Gender had no statistically significant impact on the distribution into the three groups. The mean ages were 59.52 years (SD 16.04, min 33, max 88), 55.87 years (SD 15.65, min 27, max 82) and 61.63 years (SD 18.52, min 27, 88) in the groups with likely, uncertain and unlikely NP, respectively. The was no statistically significant difference in age between the three groups. The mean follow-up PD-Q score was 15.9 (SD 7.43, min 0, max 35).

At the follow-up visit, the mean NRS was 5.0 (SD 2.67, min 0, max 10), the mean NRS of the most severe pain during the last 4 weeks was 7.16 (SD 2.23, min 0, max 10) and the mean NRS of average pain during the last 4 weeks was 5.55 (SD 2.31, min 0, max 10.0).

When we tested for a correlation between the PD-Q score and the NRS at the follow-up visit, we found a positive correlation with the PD-Q score for the current NRS score (pc = 0.3, *p* = 0.019), NRS of most severe pain during the last 4 weeks (pc = 0.34, *p* = 0.007) and NRS of average pain in the last 4 weeks (pc = 0.35, *p* = 0.005).

Equally to the statistical analysis at the baseline visit, we split the follow-up patients into PD-Q subgroups defined by likely, uncertain and unlikely NP and tested groups for correlations between PD-Q score and the NRS. In the likely NP group, we found a positive correlation with the PD-Q score for both the average NRS in the last 4 weeks (pc = 0.63, *p* = 0.003) and the current NRS at the time of the follow-up (pc = 0.52, *p* = 0.018). In the group with uncertain NP, we found no correlation between PD-Q and NRS. In the group with unlikely NP, there was a positive correlation (pc = 0.47, *p* = 0.035) between the average NRS in the last 4 weeks and the PD-Q score.

Of the 62 patients at the follow-up visit, 34 (54.8%) were treated with anti-neuropathic medication. The mean age in this group of patients was 60.56 years (SD14.17, min 32, max 88). The mean age in the patients without anti-neuropathic medication was 57.05 years (SD 19.65, min 27, max 88). There was no statistically significant difference in age between these two groups. Of the 62 patients treated with anti-neuropathic medication, 17 (50%) were women and 17 (50%9 were men. In the group without anti-neuropathic medication, 23 patients (82.1%) were women and 5 patients (17.9%) were men. We found a statistically significant (*p* = 0.008) correlation between gender and whether the patients were treated with anti-neuropathic medication or not at the follow-up visit. A subanalysis showed that 14 of the 34 patients with anti-neuropathic medication had likely NP, 14 had uncertain NP and 6 had unlikely NP.

Considering only the patients with anti-neuropathic medication, we found a positive correlation between the current NRS score (pc = 0.36, *p* = 0.039), NRS of most severe pain during the last 4 weeks (pc = 0.44, *p* = 0.011) and NRS of average pain in the last 4 weeks (pc = 0.44, *p* = 0.011) and the PD-Q score at the follow-up visit.

In the group of patients without anti-neuropathic therapy, no correlations were found between the NRS and the PD-Q score at the follow-up visit.

The mean PD-Q score in the group of patients with an anti-neuropathic therapy at the follow-up visit was 15.88 (SD 7.89, min 0, max 30), the mean NRS at the follow-up visit was 5.07 (SD 2.45, min 0, max 10), the mean NRS of average pain during the last 4 weeks was 5.97 (SD 2.36, min 0, max 10) and the NRS of the highest pain in the last 4 weeks was 7.36 (SD 2.45, min 0, max 10).

In the group without anti-neuropathic medication, the mean PD-Q score was 13.75 (SD 6, min 3, max 30), the mean NRS at the follow-up visit was 4.92 (SD 2.88, min 0, max 8), the mean NRS of average pain during the last 4 weeks was 5.05 (SD 2.19, min 0, max 8) and the NRS of the highest pain in the last 4 weeks was 6.93 (SD 1.96, min 2, max 10).

There was no significant difference in the PD-Q score between the patients with and without anti-neuropathic medication at the follow-up visit. Figure 2 shows the distribution of the PD-Q score and the NRS in patients with and without anti-neuropathic medication. The summary of all correlations found between the PD-Q score and the three NRS scores in all the defined subgroups at both time is shown in Table 2.

## 4. Discussion

Pain, particularly chronic pain, represents a major clinical, social and economic problem. Most patients with chronic pain receive multimodal treatment, partly dependent on whether pain manifests itself in a dominant neuropathic or nociceptive way. The majority of chronic pain states present as mixed ones, with a strong interindividual variability. Frequently, the diagnosis of NP is made too late or is too unspecific, leading to a delayed or inadequate pain therapy, which possibly promotes the chronification of NP. Adequate assessment by using validated tools plays an essential role in successful pain therapy.

Unidimensional pain scales such as the numeric rating scale (NRS), verbal rating scale (VRS) or visual analog scale (VAS) are useful for the measurement of pain intensity, while multidimensional scales consider multiple characteristics of chronic pain and can categorize pain more accurately. Performing the PD-Q in addition to the routine NRS scoring in an outpatient setting can help to correctly categorize the pain and thereby prevent an unnecessary delay at the beginning of a targeted anti-NP treatment if indicated.

The PD-Q was developed and validated in patients with low back pain, but it has been performed and used to estimate the prevalence of NP in several studies concerning diabetic neuropathy, postherpetic neuralgia or knee osteoarthritis [16,17].

Further studies considering NP therapy demonstrated that pain with a neuropathic component is often described in painful conditions that are not typically associated with neural lesions [8,9]. Thirty-four percent of patients with fibromyalgia reported a prevalence of NP symptoms [8]. Ohtori et al. reported that 5.4% of subjects with knee osteoarthritis had likely NP and 15.2% possibly had NP [17]. In our study, by investigating a heterogeneous group of patients with a variety of painful diseases, the proportions of subjects with likely and possible NP were higher (35.2% and 31.3%) at the first visit to the outpatient clinic.

Furthermore, the results of our study showed, according to the baseline PD-Q scores, that 66.5% of the investigated outpatients had an NP component (PD-Q score 13–38) at baseline. We demonstrated that that at the first visit to the outpatient clinic, patients were experiencing high levels of pain (NRS 6.86) accompanied with an NP component, represented in the PD-Q scores. These results suggest that a large number of chronic pain patients have poor pain control with standard analgesic regimens prior to visiting the outpatient clinic. Filtering out these patients as a distinct subgroup would allow physicians to improve their pain treatment by using anti-neuropathic analgesics with central activity. Furthermore, the PD-Q may also be a useful diagnostic tool for general physicians to screen for NP in private practice. Depending on the results of the questionnaire, and if clinically indicated, the pain therapy can be adapted immediately, or the patient can be referred to a pain specialist.

When treating chronic pain, it must be clear, that besides intensity, the type and quality of pain can change and vary over time. Therefore, repeated performance of a questionnaire, including items concerning neuropathic symptoms and intensity of pain, may be helpful to monitor the course of the overall pain and its components. An initial baseline screening followed by repeated re-evaluations of the NP component, using the PD-Q, may help physicians to guide complex pain therapy regimes.

In a study by Velluci et al. on the heterogeneity of chronic pain, it is recommended that the pain management approaches are tailored to each patient, with adequate assessment of pain by using validated tools and prompt administration of analgesics that are appropriate to the individual pain intensity [18].

However, it may be of crucial importance that in the special cohort of chronic pain patients, both the evaluation of pain intensity using the NRS score and the screening for a neuropathic component using the PD-Q are repeated at regular time intervals, regardless of whether patients have received anti-NP therapy. The PD-Q, designed primarily as a screening tool, may thus also be a helpful, easy and time-saving tool to evaluate a therapeutic outcome.

The results of our study showed a positive correlation between PD-Q scores and NRS scores measured at baseline and follow up visit. At baseline visit in patients with a PD-Q score indicating likely NP, a significant positive correlation between the NRS of average pain in the last 4 weeks and the PD-Q score was detected. At the follow-up, the correlation between PD-Q and average NRS in the last 4 weeks followed the same pattern. Analyzing the two subgroups of patients with and without an anti-NP therapy, the correlations between PD-Q and NRS scores changed in opposite directions to each other at both points in time. In the anti-NP therapy group, there was no correlation at baseline visit but there was a correlation at follow-up between PD-Q and NRS, and vice versa in the subgroup of patients without anti-NP therapy. Furthermore, there was a significant difference in the PD-Q score between the patients with and without anti-neuropathic medication at baseline visit but none at the follow-up visit.

However, based on these results, it can be assumed that the partly controversial trend in the subgroups is due to the anti-NP therapy. This trend is reinforced by the results showing that the correlations in the PD-Q subgroups of likely NP and unlikely NP change with similar tendency, which may be attributed to the anti-neuropathic therapy. Further, our results indicate that a large number of chronic pain patients have likely or possible NP at their first visit to the outpatient pain clinic. The implementation and re-evaluation of a simple NP questionnaire like the PD-Q in routine clinical practice would contribute to early recognition and appropriate treatment of these patients. All these study findings point out that the relevant factor in this context is the anti-neuropathic medication. A potential limitation of this study was the relatively small number of patients in the follow-up and in the subgroups of patients with and without anti-NP therapy; therefore, further studies concerning this special topic with longer follow-up intervals are warranted. They should contribute in determining whether improvement over time is observed in subjects with anti-neuropathic therapy and PD-Q scores indicating likely NP.

## 5. Conclusions

The results of this study show a significant correlation between PD-Q score and NRS score at baseline and 1-month follow-up visit. These results demonstrate that the neuropathic component is closely related to the experience of intense pain sensations, represented in both the PD-Q and NRS scores. However, this study highlights the importance of implementing a validated questionnaire like the PD-Q in the daily routine of pain therapy to screen for a neuropathic component in a wide range of chronic pain diseases.

## Figures and Tables

**Figure 1 medicina-57-00353-f001:**
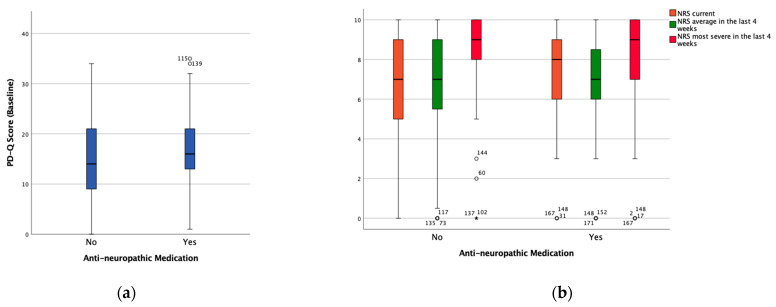
Boxplots showing the distribution of (**a**) the PainDETECT questionnaire (PD-Q) score and (**b**) the numeric rating scale (NRS) in the groups with (Yes) and without (No) anti-neuropathic medication at the baseline visit. ° and * represent outliers and extreme outliers respectively.

**Figure 2 medicina-57-00353-f002:**
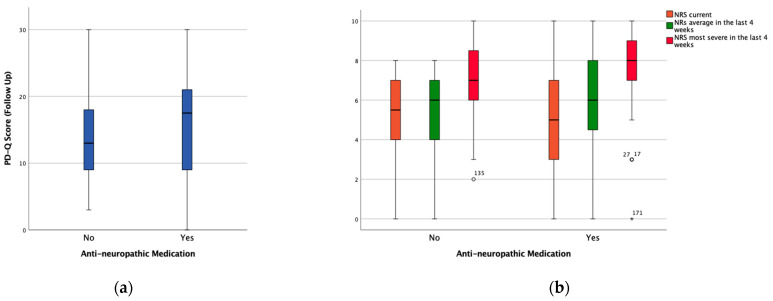
Boxplots showing the distribution of (**a**) the PD-Q score and (**b**) the NRS in the groups with (Yes) and without (No) anti-neuropathic medication at the baseline visit. ° and * represent outliers and extreme outliers respectively.

**Table 1 medicina-57-00353-t001:** Patient distribution by ICD code.

ICD-10 Code	Description	Number of Patients
B00–B09	Viral infections characterized by skin and mucous membrane lesions	5 (2.8%)
G40–G47	Episodic and paroxysmal disorders	20 (11.4%)
G50–G59	Nerve, nerve root and plexus disorders	17 (9.7%)
G60–G64	Polyneuropathies and other disorders of the peripheral nervous system	4 (2.3%)
G90–G99	Other disorders of the nervous system	5 (2.8%)
L80–L99	Other disorders of the skin and subcutaneous tissue	2 (1.1%)
M00–M25	Arthropathies	21 (11.9%)
M40–M54	Dorsopathies	82 (46.6%)
M60–M79	Soft tissue disorders	13 (7.4%)
M95–M99	Other disorders of the musculoskeletal system and connective tissue	4 (2.3%)
R10–R19	Symptoms and signs involving the digestive system and abdomen	4 (2.3%)
R50–R69	General symptoms and signs	18 (10.2%)

**Table 2 medicina-57-00353-t002:** Summary of all correlations found between the PD-Q score and the three numerical rating scales. NRS scores in all the defined subgroups at both time points in the study. Neuropathic pain (NP).

	Correlation with PD-Q Score	
	Baseline Visit	Follow-Up Visit
Overall		
NRS current	pc = 0.21, *p* = 0.007	pc = 0.3, *p* = 0.019
NRS most severe in the last 4 weeks	pc = 0.2, *p* = 0.01	pc = 0.34, *p* = 0.007
NRS average in the last 4 weeks	pc = 0.26, *p* < 0.001	pc = 0.35, *p* = 0.005
Likely NP		
NRS current	-	pc = 0.52, *p* = 0.018
NRS most severe in the last 4 weeks	-	-
NRS average in the last 4 weeks	pc = 0.4, *p* = 0.01	pc = 0.63, *p* = 0.003
Uncertain NP		
NRS current	-	-
NRS most severe in the last 4 weeks	-	-
NRS average in the last 4 weeks	-	-
Unlikely NP		
NRS current	-	-
NRS most severe in the last 4 weeks	-	-
NRS average in the last 4 weeks	-	pc = 0.47, *p* = 0.035
With anti-neuropathic medication		
NRS current	-	pc = 0.36, *p* = 0.039
NRS most severe in the last 4 weeks	-	pc = 0.44, *p* = 0.011
NRS average in the last 4 weeks	-	pc = 0.44, *p* = 0.011
Without anti-neuropathic medication		
NRS current	pc = 0.21, *p* = 0.039	-
NRS most severe in the last 4 weeks	pc = 0.26, *p* = 0.012	-
NRS average in the last 4 weeks	pc = 0.3, *p* = 0.003	-

## Data Availability

All data to support the findings of this study are available contacting the corresponding author upon request.

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
