# Peer review of "Correlation between the Neuropathic PainDETECT Screening Questionnaire and Pain Intensity in Chronic Pain Patients"

_medicina, 2021, doi:10.3390/medicina57040353_

Round 1

Reviewer 1 Report

Dear Authors,

The paper is overall well structured despite a some comments. I have suggestion that could improve the paper:

  • Introduction
    • The introduction explains the most important information about the study.
  • Material and methods:
    • The NRS scale is an ordinal scale, explain, why was the t test used. In my opinion would be better use Mann-Whitney U test or Wilcoxon test.
    • Explain, why the included patients had various underlying diagnoses.
  • Results:
    • Present the characteristics of the group (e.g. age, gender, etc.)
    • Present whether there were statistically significant differences between the groups (e.g. age, gender, etc.). If yes, then do your analysis for confounding factors.
    • It would be better describe the results in the table.
  • Discussion
    • In my opinion the discussion is well written.
  • Conclusion
    • On conclusion you written about wide range of chronic pain diseases. But you studied only 4 patients with symptoms and signs involving the digestive system and abdomen and other disorders of the skin and subcutaneous tissue. In my opinion, it would be better in your research to be limited to one group of diseases, e.g. cardiological, neurological, etc.

Author Response

Response to Review 1

Dear Reviewer, thank you for your recommendations.

  1. Introduction: The introduction explains the most important information about the study.
  2. Material and methods:
    • The NRS scale is an ordinal scale, explain, why was the t test used. In my opinion would be better use Mann-Whitney U test or Wilcoxon test.

Response: We applied the Mann-Whitney U test for independent variables as you suggested. (see 120, 189, 251)

  • Explain, why the included patients had various underlying diagnoses.

Response: The patients were recruited based on their first visit to our pain clinic. We aimed to investigate chronic pain patients with different pain diagnoses to demonstrate that the NP component can play a major part in chronic pain sensations, where neuropathic pain is not typically associated with.

Our study results showed, according to the baseline PD-Q scores, that 66.5 % of the investigated outpatients, with different painful diseases, had a NP component at their first visit to the pain clinic. These results show that patients were experiencing high levels of pain (NRS 6.86) accompanied with a NP component, represented in the PD-Q scores. These study results suggest that chronic pain patients at their first visit to the pain clinic, have poor pain control with standard analgesic regimens, despite years of different therapy approaches. So detecting these special cohort of patients, would allow physicians to improve their pain treatment by using anti-neuropathic analgesics in an early stage of treatment.  So our data, obtained in a group of chronic pain patients with mixed pain diagnoses, may be the basis for further research with the focus on one of these subgroups analyzed.

  • Results:
    • Present the characteristics of the group (e.g. age, gender, etc.)

Response: We have supplemented the group characteristics of the analysed subgroups, taking into account the gender and age aspect. (see 146, 149, 167, 174, 200, 225)

    • Present whether there were statistically significant differences between the groups (e.g. age, gender, etc.). If yes, then do your analysis for confounding factors.

Response: There were no statistically significant differences between the groups with respect to gender and age, thus not emphasizing these.

    • It would be better describe the results in the table.

Response: We gratefully accept the advice and have presented study results in a table.

  • Discussion
    • In my opinion the discussion is well written.

Thank you for the positive feedback.

  • Conclusion
    • On conclusion you written about wide range of chronic pain diseases. But you studied only 4 patients with symptoms and signs involving the digestive system and abdomen and other disorders of the skin and subcutaneous tissue. In my opinion, it would be better in your research to be limited to one group of diseases, e.g. cardiological, neurological, etc.

Response: Chronic pain with a neuropathic component is often described in painful conditions which are not typically associated with a neural lesions or neuropathic pain. Previous projects have already studied patients “limited to one group of painful diseases” such as from French HP. or Valdes and colleagues (see references). We aimed to “screen” all chronic pain patients with mixed diagnoses in painful conditions at their first visit to our pain clinic. The NP component can play a major part in chronic pain sensations. This study highlights the importance of implementing a validated questionnaire like the PD-Q in daily routine of chronic pain therapy to screen for a neuropathic component.  Our study results showed, according to the baseline PD-Q scores, that 66.5 % of the investigated outpatients had a NP component at baseline visit, demonstrating that these patients have poor pain control with standard analgesic regimens, despite years of different therapy approaches. So detecting these special cohort of patients, would allow physicians to improve their pain treatment by using anti-neuropathic analgesics in an early stage of treatment.  The implementation and re-evaluation of a simple NP questionnaire like the PD-Q in routine clinical practice regardless of primary pain-associated diagnoses, would contribute to early recognition and appropriate treatment. So depending on the questionnaires results pain therapy can be adapted immediately or the patient referred to a pain specialist.

Reviewer 2 Report

This study results shows a significant correlation between PD-Q score and NRS score at baseline and 1-month follow up visit. The authors also demonstrate that the neuropathic component is closely releated to the experience in intense pain sensations, represented  in both, the PD-Q and NRS score. However, this study highlights the importance of implementing a validated questionnaire like the PD-Q in daily routine of pain therapy to 
screen for a neuropathic component in a wide range of chronic pain diseases.

The methodology is well written and the design is appropriate . Excluding the statistical analysis ( which is not my expertise) the results do make sense.

The manuscript must be revised for language and grammar other wise is acceptable if editors find the statistical analysis appropriate. 

Author Response

Reviewer 2:

This study results shows a significant correlation between PD-Q score and NRS score at baseline and 1-month follow up visit. The authors also demonstrate that the neuropathic component is closely related to the experience in intense pain sensations, represented  in both, the PD-Q and NRS score. However, this study highlights the importance of implementing a validated questionnaire like the PD-Q in daily routine of pain therapy to 
screen for a neuropathic component in a wide range of chronic pain diseases.

The methodology is well written and the design is appropriate. Excluding the statistical analysis (which is not my expertise) the results do make sense.

The manuscript must be revised for language and grammar otherwise is acceptable if editors find the statistical analysis appropriate. 

Response : Thank you for the positive feedback. The manuscript was revised for language and grammar as recommended.

Round 2

Reviewer 1 Report

Dear Authors,

Thank you for your corrections.